# On the Homology of the Dominant and Non-Dominant Corticospinal Tracts: A Novel Neurophysiological Assessment

**DOI:** 10.3390/brainsci13020278

**Published:** 2023-02-07

**Authors:** Maria Rita Pagliara, Federico Cecconi, Patrizio Pasqualetti, Massimo Bertoli, Karolina Armonaite, Eugenia Gianni, Joy Grifoni, Teresa L’Abbate, Franco Marinozzi, Livio Conti, Luca Paulon, Antonino Uncini, Filippo Zappasodi, Franca Tecchio

**Affiliations:** 1LET’S and LABSS, Institute of Cognitive Sciences and Technologies (ISTC), National Research Council of Italy (CNR), 00185 Rome, Italy; 2Department of Mechanical and Aerospace Engineering, Sapienza University of Rome, 00185 Rome, Italy; 3Department of Public Health and Infectious Diseases, Sapienza University of Rome, 00185 Rome, Italy; 4Department of Neuroscience, Imaging and Clinical Sciences, University “G. d’Annunzio” of Chieti-Pescara, 66100 Chieti, Italy; 5Uninettuno University, 00186 Rome, Italy; 6Unit of Neurology, Neurophysiology, Neurobiology, Department of Medicine, University Campus Bio-Medico di Roma, 00128 Rome, Italy; 7Istituto Nazionale di Fisica Nucleare, Sezione Roma Tor Vergata, 00133 Rome, Italy; 8Engineer Freelance, 00159 Rome, Italy

**Keywords:** corticospinal tract, novel-concept physiological measures, hemi-body homology, on-center off-surround, handedness

## Abstract

Objectives: The homology of hemispheric cortical areas plays a crucial role in brain functionality. Here, we extend this concept to the homology of the dominant and non-dominant hemi-bodies, investigating the relationship of the two corticospinal tracts (CSTs). The evoked responses provide an estimate of the number of in-phase recruitments via their amplitude as a suitable indicator of the neuronal projections’ integrity. An innovative concept derived from experience in the somatosensory system is that their morphology reflects the recruitment pattern of the whole circuit. Methods: CST homology was assessed via the Fréchet distance between the morphologies of motor-evoked potentials (MEPs) using a transcranial magnetic stimulation (TMS) in the homologous left- and right-hand first dorsal interosseous muscles of 40 healthy volunteers (HVs). We tested the working hypothesis that the inter-side Fréchet distance was higher than the two intra-side distances. Results: In addition to a clear confirmation of the working hypothesis (*p* < 0.0001 for both hemi-bodies) verified in all single subjects, we observed that the intra-side Fréchet distance was higher for the dominant than the non-dominant one. Interhemispheric morphology similarity increased with right-handedness prevalence (*p* = 0.004). Conclusions: The newly introduced measure of circuit recruitment patterning represents a potential benchmark for the evaluation of inter-lateral mechanisms expressing the relationship between homologous hemilateral structures subtending learning and suggests that variability in recruitment patterning physiologically increases in circuits expressing greater functionality.

## 1. Introduction

The dynamic interaction between the activities of homologous cortical areas is a key determinant of proper brain function. In fact, behavioral performance depends on the functional connectivity between homologous hemilateral nodes of dedicated networks even at rest [1,2]. Hemispheric connections are supported by corpus callosum fibers, which bi-directionally connect left and right homologous cortical parcels [3].

Focusing on the right and left primary motor areas (M1), trans-callosal fibers have been shown to mediate interhemispheric interaction primarily with inhibitory effects [4,5,6,7]. While a small portion of callosal inputs on pyramidal neurons exerts direct GABAergic inhibitory modulation, most interhemispheric trans-callosal fibers are excitatory glutamatergic projections acting on contralateral inhibitory networks [4]. In fact, the action of an M1 on the homolog is largely conditioned by local inter-neural interactions mediated by low-threshold cortical inhibitory neurons [5,6].

These cortical inhibition mechanisms [7,8] are key elements in the acquisition of skillful motor control and sensory perceptual acuity [9] and are implemented using lateral surround inhibition processes, called center on surround off. The balance between the activities of homologous hemispheric areas, implemented using local inhibitory mechanisms common to the processing capacity of fine neural networks, is thus a ubiquitous structural–functional mechanism that supports the brain’s plastic adaptation and learning processes [10,11]. In other words, the interaction between homologous structures in the two hemispheres is an integral and critical part of the inhibitory–excitatory functional circuits that support the functionality of the body segment they control.

Beyond the balance between the two hemispheres, here, we exploit muscle responses obtained using transcranial magnetic stimulation (TMS), i.e., motor-evoked potentials (MEPs), to noninvasively investigate the balance between homologous center–peripheral segments.

The physiological function of the corticospinal tract (CST) is related to the dynamic interaction between different intra- and interhemispheric cortical brain regions and can be assessed under both normal and pathological conditions by using the single-pulse supra-threshold TMS on M1 [12] that propagates through the spinal cord and can be de-treated at the contralateral muscle level with surface electrodes as an MEP [13].

An idea for assessing the relationship between hemilateral homologues by exploiting the morphology of evoked responses as an index of the recruitment patterns of the circuit involved can be derived from studies of somatosensory evoked potentials (SEPs) and fields (SEFs). Indeed, albeit with high inter-individual variability, a similar morphology of the somatosensory evoked responses in the right and left hemisphere reflects similar patterns of the activation of the homologous peripheral-central pathways [14]. Moreover, even within the same system, districts with common innervation generate responses with more similar morphologies than the districts supplied by different nerves [15].

Similarly, here, we aimed to quantify the balance physiology of CSTs in the two sides of the body by assessing the similarity of the morphology of left and right MEPs in healthy volunteers.

For this purpose, we used Fréchet distance [16,17], a measure of similarity between the shapes of two curves, commonly used in geographic information systems technology to compare the similarity of path shapes. The Fréchet distance is zero if the two curves are equal and increases positively as the curves become more dissimilar. Applied to biological data, it has already been shown to recognize spatiotemporal networks between groups of subjects from functional magnetic resonance imaging (fMRI) [18], as well as the spatiotemporal surface atlases of the developing cerebral cortex [19]. In addition, Fréchet distance has recently enabled the sensitive classification of post-stroke motor recovery levels by quantifying inter-patient similarity between post-lesion corticospinal tract fractional anisotropy profiles [20].

To introduce a measure of “physiological” recruitments, we considered the enormous variability typical of neuronal responses to any sensory input or motor execution. In the present study, we directly exploit the variability of an MEP, well known to reach 80% of its mean value when estimated from amplitude. We test the working hypothesis that the similarity of the morphology of the inter-lateral MEPs is less than that of intra-lateral MEPs. Through meeting this criterion, we aim to support the reliability of the measure and simultaneously provide its reference range for homology between the two corticospinal tracts under physiological conditions.

## 2. Methods

The study was conducted in accordance with the Declaration of Helsinki and de-written according to the Recommendations of the International Committee of Medical Journal Editors (IC-MJE). It was approved by the competent Ethics Committee. All subjects signed informed consent before enrollment.

### 2.1. Study Design

The present investigation focuses on the similarity of inter-lateral MEP morphology as an ex-prime of the balance of the two homologous corticospinal tracts. We designed the analysis to test the hypothesis that inter-lateral similarity is lower than intra-lateral similarity for both dominant and non-dominant CSTs.

### 2.2. Healthy Volunteers’ Population

Forty people were enrolled. Health status was assessed through clinical history, considering exclusion criteria including history of seizures, psychiatric or neurological disorders, implants, and the taking of medications that may modify cortical excitability. Hand dominance was tested through the Edinburgh Handedness Inventory score [21].

### 2.3. MEP Collection and Analysis

#### 2.3.1. Stimulation and Recording Setup

Subjects were seated in a comfortable chair in a quiet room (Figure 1). Muscle signals (electromyogram, EMG) of the first dorsal interosseous muscle (FDI) of the right and left hand were detected by two surface electrodes in a belly–tendon assembly (2.5 cm apart). We performed single-pulse TMS through a standard focal coil connected to a Bistim 200 module (The Magstim Company Ltd., Whitland, UK). For each subject, we searched for the coil position that evokes optimal MEP from the contralateral IDE (hotspot position) and assessed the resting motor threshold (RMT), defined according to international standards as the intensity that elicits MEP on the 50-microV amplitude scale in approximately 50% of 16 consecutive trials (Rossini et al. 1994). TMS was applied at an intensity adjusted to 120% of the RMT. TMS stimuli were elicited with an inter-pulse interval of 5 s, collecting 20 repetitions on each side [22,23], first in the left and then in the right hemisphere. Subjects were asked to remain still and relaxed for two to three minutes during data collection for each lateral stimulation. EMG activity was collected continuously throughout the stimulation to reject traces with detectable activity before the stimulus, visible at 50 μV amplification.

#### 2.3.2. MEP Morphology Similarity Estimate

We used the Fréchet distance estimate implemented in Matlab [24]. The Fréchet distance between two curves estimates the minimum bead length sufficient to join a point traveling forward along one of the two curves and a point traveling forward along the second, with the travel speed for both points not constrained to be uniform, and which is optimized by the algorithm.

To estimate the individual similarity between the two homologous CST hemi-bodies in each subject, we estimated the Fréchet distances (Figure 1) between each of the 20 left (sn) and each of the 20 right (dx) FDI MEPs in a time window from 15 ms to 45 ms (0 being the arrival of the TMS pulse at the cortical hotspot), yielding 400 left-right (DxSn) Fréchet MEP distances. The intra-sided estimates correspondingly consisted of $\left(\begin{array}{c} n\\ k \end{array}\right)$ with n = 20 and k = 2, resulting in 190 Fréchet distances (for each DxDx and SnSn).

#### 2.3.3. MEP Amplitude Estimate

For comparison with the well-established standard analysis, assessed via MEP latency and amplitude, we calculated the mean values. To evaluate whether the MEP amplitude can provide a simple method to assess inter-side balance, we also tested the working hypotheses that the inter-side MEP amplitude difference variability was higher that both the right and left intra-side ones.

#### 2.3.4. Statistical Analysis

To check the normality of the distributions, we did not rely on formal statistical tests (such as Shapiro–Wilk), because, for hundreds of values, these tests are too sensitive, and even small deviations from Gaussianity are statistically significant. Instead, we verified that, in each subject, the distribution was mostly right-handed and approximately log-normal, therefore we applied the log-transform to the original Fréchet distance measures. In almost every subject we obtained an improved Gaussian fit and a substantial reduction in outliers, as confirmed by a visual inspection of the quantile–quantile (Q-Q) plots before and after log-transformation.

To investigate the morphological similarities for inter-sided and intra-sided comparisons, a linear mixed model was applied to the single variable Fréchet distance, considering the within-subject factor *Morphology Similarity* (DxSn, DxDx, SnSn) and the randomfactor *Healthy Volunteer* (HV1, HV2, …, HV40). If the *Morphology Similarity* factor was significant, post-hoc comparisons (with Sidak correction) were performed to check the differences of each intra-side with the inter-side comparison and whether a difference emerged between the intra-side morphology within dominant and non-dominant sides. When *Sex* was added in the model as an additional source of variation, the random-factor *Healthy Volunteer* was entered as nested in sex and the effect of sex was assessed both as a main and interactive *Sex*Morphology Similarity* factor.

We reported a result for the significance of the effect *p* < 0.050.

To be consistent with the current literature, we also evaluated the latency and amplitude of MEPs in the two sides of the body and the inter-lateral difference, focusing particularly on the variance of intra- and inter-lateral amplitude, again testing the hypothesis that inter-lateral variability was higher than intra-lateral variability.

Statistical analysis was performed with SPSS 27.

#### 2.3.5. Data Availability

MEPs, Fréchet algorithms, and personal and clinical anonymized data will be available upon request.

## 3. Results

### 3.1. Healthy Volunteers’ Population and MEP Features

The 40 individuals (21 females and 19 males) had a mean age of 24.8 years (sd 3.7, range [20,25]). Their TMS resting motor threshold was highly comparable in the two hemispheres, and the same was true for latency and amplitude (Table 1). The mean of the Edinburgh Handedness Inventory (EHI), collected in a subgroup of 22 subjects, was 61.2 ± 31.9, with a range of [−30, 100]. Considering that left-handedness is indicated by EHI ≤ 40, ambidextrousness by −40 < EHI < 40, and right-handedness by EHI > 40, the subpopulation included no left-handed, 3 ambidextrous, and 19 right-handed.

### 3.2. CTS Homology via MEP’s Fréchet Distance

Linear mixed model indicated a strong *Morphology Similarity* effect [F(2,78.0) = 68.9, *p* < 0.001], emerging even though a clear *Healthy Volunteer*Morphology Similarity* interaction effect was found [F(78,30047) = 72.7, *p* < 0.0001]. Since the *Morphology Similarity* factor was significant, we estimated the three pairwise differences and found a clear higher Fréchet distance for the DxSn comparison with respect to both the DxDx (log unit: 0.343, Sidak adjusted 95% CI: 0.245, 0.441; *p* < 0.001) and the SnSn (log unit: 0.432, Sidak adjusted 95% CI: 0.334, 0.530; *p* < 0.001) (Figure 2). When comparing the intra-side morphology similarity, we found that the dominant DxDx Fréchet distance was higher than the non-dominant SnSn, although the null hypothesis of their similarity could not be rejected at 0.05 alpha level (difference in log unit: 0.089, Sidak adjusted 95% CI: −0.025; 0.203, *p* = 0.063). Notably, the comparison differed strongly among people as evidenced by the *Subject*Morphology Similarity* interaction effect [F(78,30047) = 72.67, *p* < 0.0001]. Investigating the 190 Fréchet distance estimates in each condition for each person, we observed in single subjects that in 22 DxDx > SnSn (*p* < 0.001 consistently); the opposite occurred in 12 and non-significant difference appeared in 6 subjects.

When the possible role of Sex was taken into consideration, the linear mixed model confirmed the *Morphology Similarity* effect [F(2,76) = 67.2, *p* < 0.001], and a *Sex* effect on the Fréchet distance was also found [F(1,38) = 4.88, *p* = 0.033; Figure 3], due to higher heterogeneity in females (difference of log Fréchet distance: 0.179, 95% CI: 0.11, 0.245). No interaction *Sex* * *Morphology Similarity* was found [F(2,38) = 1.71, *p* = 0.820].

The inter-side morphology similarity was related to handedness, with a Pearson’s rho = −0.585, *p* = 0.004 (on the 22 people from whom we collected the Edinburgh Handedness Inventory test).

### 3.3. CTS Homology via MEP’s Amplitudes

To study the MEP amplitude, we applied the logarithmic transformation, obtaining a good fit of the normal distribution. The variances of the amplitude differences, when comparing the left- and right-sided MEPs, did not differ from the intra-sided variances (Figure 4). The intra-side amplitude difference had a mean variance of 0.21, while the mean intra-side variance was 0.27 for the right FDIMEP and 0.27 for the left FDIMEP. The paired-samples t-test between inter-lateral and intra-lateral variance had *p* = 0.702 with the right and *p* = 0.312 with the left. Moreover, no effect was found when comparing the right and left intra-lateral variances (*p* = 0.709).

## 4. Discussion

We obtained two key results. According to the working hypothesis, the morphology of the CST-evoked response is less similar between the two homologous hemi-corps than within each of them. In contrast, the second key result of our investigation is quite unexpected, as it concerns an overall less similar response from the dominant side than from the non-dominant side.

### 4.1. Higher Variability of Recruitment Pattern in the Dominant than the Non-Dominant Hemibody

Neural network physiology is based on an intimate variability [26] of the “repetitive events” that occur in response to environmental sensory input, internal physiological processing, and behavioral action realization [27]. The origin of this intimate variability [28] emerges from the processing of neuronal units, where the continuous integration of incoming projections across the large-scale anatomical structure of the human brain generates a self-regulated timing of activation; moreover, at multiple scales, the overall exchanges between neuronal pools even in the resting state show variability within and between temporal occurrences and state changes [29]. A network can exhibit a linear response, despite the highly nonlinear dynamics of individual neurons, and react to changing external stimuli on time scales much smaller than the integration time constant of a single neuron [30,31]. The case of TMS-evoked responses is paradigmatic, with an inter-trial variability of about 80 percent [32,33].

Greater variability in the recruitment pattern of the dominant CST may emerge as an effect of more refined local circuits developed by learning in the dominant side that support a broader and more flexible behavioral repertoire. The same circuits that express the center on surround off circuiting mechanism that encodes learning perception and fine control by the dominant hand [34] are the circuits onto which the fibers of the homologous cortical area of the other hemisphere project. We may assume that the variability in CST recruitment expresses greater executive capacity of the dominant side. We believe that the local circuits of the origin of the fibers of the dominant corticospinal tract express in the variability of recruitment an effect of structural differences, where the areas of the dominant central sulcus are deeper and have greater connectivity than the non-dominant ones [35].

### 4.2. CST Homology vs. Intra-Side Recruitment Pattern Similarity

Here, the investigation of local and homologous structures aims to reveal a simple measure of the crucial phenomenon underlying learning, namely the construction of center on surround off networks that enable the specific acquisition of motor control [36] and sensory processing [37] through a continuous sensory–motor neuronal–mechanical feedback interaction [25,38]. In support of a common mechanism of local learning and interaction between homologous areas, we note that the correlation between similarities in recruitment patterns is stronger between the homologous and the dominant side (DxSn with DxDx Pearson correlation r = 0.74) than between the homologous and the non-dominant side (DxSn with SnSn r = 0.57) or between the two intra-sides (DxDx with SnSn r = 0.54).

By estimating the relationship between homologous hemi-corporeal structures through the morphology of the evoked response, we emphasize the relevance of network-circuit properties in neuronal communication. We have developed a rather extensive experience in assessing the homology of cortico–subcortical structures in somatosensory information processing [14,15,39,40], and have recently implemented a new method of assessing shape similarity, which has been shown to be more sensitive to physiological changes [40], in agreement with previous in vitro neuronal network studies [41].

### 4.3. Sex Effect on CST Recruitment Variability

We found that males had a smaller Fréchet distance than females and were overall intra-sided and more clearly in the inter-sided comparison. In other words, males had homologous MEP morphologies that varied less than females, and this effect was also present intra-side overall. Notably, in the same vein, not only did females tend to have greater variability in MEPs than males [42], but signs of a sex-linked difference emerged in the organization of relations between homologous areas, with females showing greater trans-callosal inhibition than males, suggesting gender differences in inhibition–excitation balances modulated by interhemispheric connectivity at least for the section of the corpus callosum devoted to the sensorimotor regions [43]. In agreement with the hypothesis of females’ cortices characterized by more efficient inhibitory networks, data from neuromodulation interventions showed women prolonged the inhibitory aftereffects of cathodal transcranial direct current stimulation than males [44].

### 4.4. Future Neuroscience and Clinical Implications

Neuroscience has developed a clear notion that the balances between homologous structures in our bodies and brains are crucial for physiological well-being. For this reason, the development of simply accessible measures of these balances is important from a clinical perspective for prognostic indications, to tailor therapeutic interventions, and to monitor the effects of rehabilitation. In our experience, we observed that this measure provided a deeper understanding of the brain–body level effects underlying fatigue relief in multiple sclerosis patients through personalized neuro-modulation intervention (data in publication). Furthermore, in the neuroscience dimension, we believe the present approach can enhance the progression of neuro-inspired artificial intelligence (AI) models. The neuronal mechanism of center on surround off information encoding, which develops in conjunction with corpus callosum-mediated interaction between homologous hemilateral areas, will be crucial for more effective modeling, which can be profitably focused on database platforms such as the Virtual Brain integrated in Ebrains.

### 4.5. MEP Shape vs. Amplitude

We found that an assessment—via Fréchet distance—of MEP morphology was able to quantify the inter-lateral and intra-lateral similarity of CST patterning across MEP morphology. Moreover, MEP amplitudes emerged with similar variability for inter-lateral and intra-lateral comparison in dominant and non-dominant corticospinal tracts.

Overall, and in agreement with recent large-scale assessments of the morphology of MEPs [45], the results discussed here are consistent with the emergence of new measures that, although in a limited sample, are consistent in indicating the morphology of the evoked response as sensitive and capable of providing new information about the network nature of the circuits involved.

### 4.6. Limitations of the Present Work

The present exploratory work has limitations, which could be addressed by future research.

Homology of CTS was observed in relation to intra-lateral similarity, and integration of this measure with behavioral characteristics would be very interesting.

We analyzed CST by comparing dominant and non-dominant hemi-corps with the level of handedness, which showed a clear association between hand dominance and inter-lateral homology, only in a subset of the study population.

We collected the MEP in blocks, first in the left hemisphere and then in the right hemisphere. In the future, random distribution in the two sides should be implemented in the experimental design.

Our investigation was performed by exploiting a hand muscle in a highly homogeneous age group. The stability of the results presented in different involved districts, or the age dependence, may be the subject of future investigations.

## 5. Conclusions

Our investigation reveals the chosen measure of MEP shape similarity to be appropriate for marking patterns of corticospinal tract recruitment, as it is sensitive to greater intra- than inter-lateral similarity.

## Figures and Tables

**Figure 1 brainsci-13-00278-f001:**
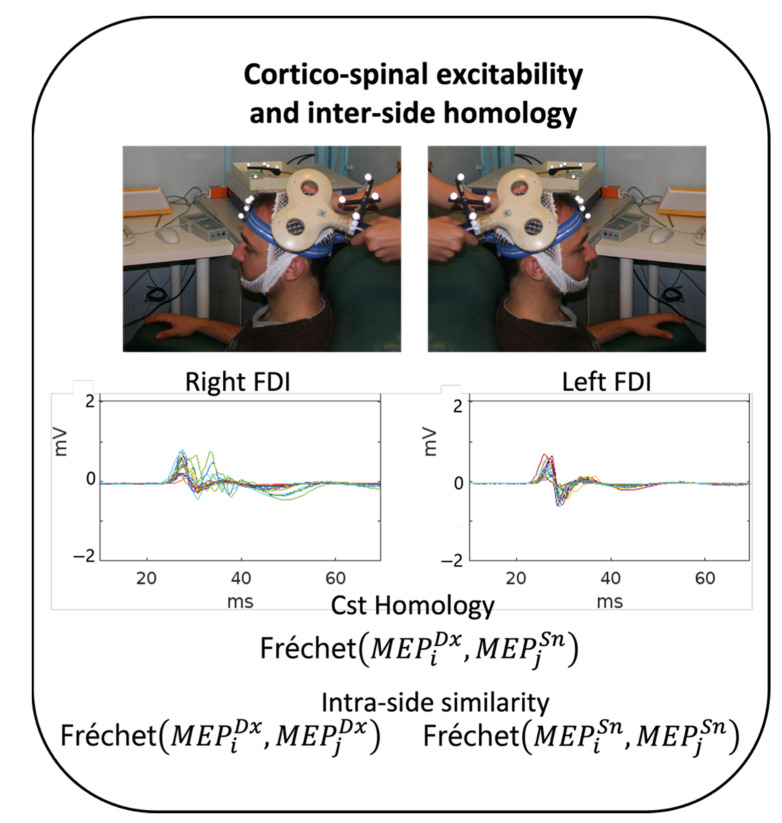
Experimental setting. Experimental procedure in each healthy volunteer (HV). Top: In each hemisphere, once the TMS FDI hotspot was identified, the position of the coil was monitored using a neuronavigational system. Approximately 20 MEPs were collected from each side, shown here superimposed in the 0–70 ms window of a representative HV. Bottom: CST homology was estimated as Fréchet distances between each pair of left and right MEP (homologous) morphologies (400 values in each subject) and between each pair of MEPs from the same side (190 values for each subject and each side).

**Figure 2 brainsci-13-00278-f002:**
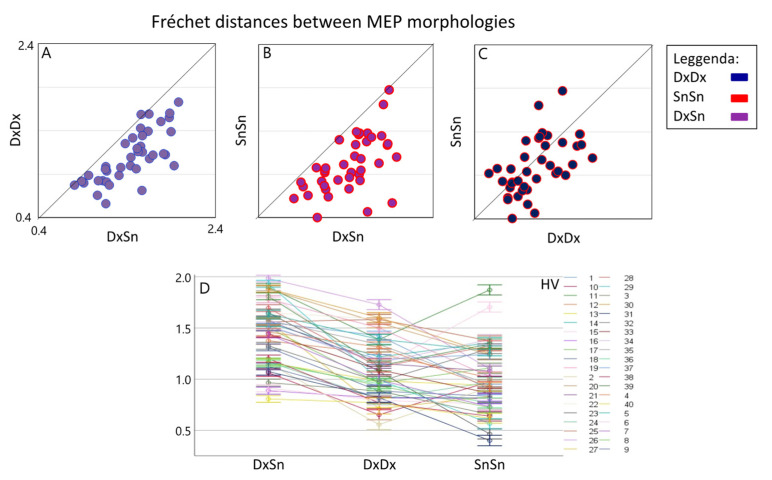
Inter-side and intra-side MEP morphology similarity. Boxplots of individual Fréchet distances between inter-lateral MEP morphologies (DxSn) (average of about 400 pairs in each HV, *x*-axis) versus intra-lateral distances (average of about 190 values). It is evident that the inter-lateral distance is higher than both the right intra-lateral distance (**A**) and the left intra-lateral distance (**B**) in all subjects. In the right intra-side comparison, a prevalence of greater distances emerges compared to the left (**C**). The overall values and standard deviation with subjects are shown in (**D**).

**Figure 3 brainsci-13-00278-f003:**
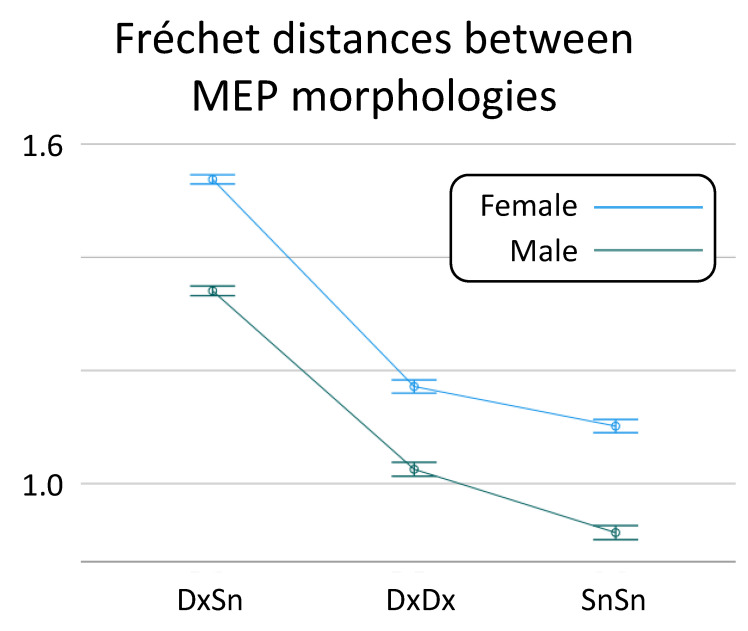
Sex-dependent MEP morphology similarity. The overall values and standard deviation of Fréchet distances between inter-side (DxSn) MEP morphologies, the intra-side distances in the right (DxDx) and left (SnSn) distinct for Sex.

**Figure 4 brainsci-13-00278-f004:**
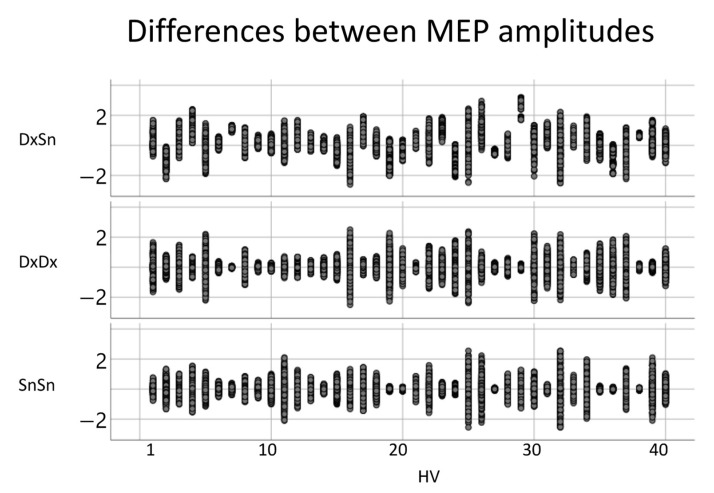
Inter-side and intra-side MEP amplitudes. For each healthy volunteer (HV), the difference between each pair of MEP amplitudes for left-right comparison (first row) and intra-side comparison (right side, second row; left third row).

**Table 1 brainsci-13-00278-t001:** TMS-estimates excitability features.

RMT (%)			MEP			
Dx	Sn	Delta	Lat Dx	Lat Sn	Amp Dx	Amp Sn
48.1	48.1	−0.1	26.3	26.4	1.7	1.3
9.1	8.1	3.3	0.9	1.2	0.6	0.5

Mean and *standard deviation* of resting motor threshold (RMT) assessed by TMS, expressed as a percentage of maximum stimulator power (%), and latency (Lat, ms) and amplitude (Amp, mV) of motor-evoked potentials (MEPs) for right (Dx) and left (Sn) first dorsal interosseous muscles. For the amplitudes, the averages were estimated after logarithmic transformation, and the inversely exponentially transformed average is presented here.

## Data Availability

TMS data will be made available upon request.

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
