# Peer review of "On the Homology of the Dominant and Non-Dominant Corticospinal Tracts: A Novel Neurophysiological Assessment"

_brainsci, 2023, doi:10.3390/brainsci13020278_

Round 1

Reviewer 1 Report

Comments and Suggestions for Authors

This is a very important fundamental work that aims to investigate the homology of the dominant and non-dominant cortico-spinal tracts. I have few suggestions on improving this manuscript:

1.       There are lateral and anterior cortico-spinal tracts. Please specify which one is investigated in this work.

2.       How many female participants are involved, and any sex-difference has been detected in the results?

3.       What are the clinical or neuroscience implications of the findings?

Reviewer 2 Report

Comments and Suggestions for Authors

This paper reports an investigation into the similarity in form, latency and amplitude of the MEPs evoked from human M1 using TMS, and recorded from an intrinsic hand muscle (FDI).  The main idea is to compare successive MEPs recorded from the same muscle (intra-muscular) in one hand with those in the other hand and evoked by TMS delivered to the opposite hemisphere. The novelty of the approach is to use the Frechet distance to compare the form of MEPs. 40 young volunteers were investigated and most of these were right-handers, with a few ambidextrous subjects. There are two main new findings: first that the variability of the form of MEPs between muscles was higher than within a muscle, and second that variability of MEPs was higher in the FDI of the dominant hand than in the non-dominant.

The first of these results is perhaps unsurprising; the second is interesting but difficult to interpret. My general comment is that the authors make too big a leap between their results, which after all reflect a motor response to a very unnatural stimulus, with complex functions such as motor learning. Readers would be helped if rather than just give the Frechet distance comparisons as statistics, please explain any possible biological significance of the scale of differences discovered.  

Detailed comments

1.       The authors do not mention that the MEP is a muscular and not a neural response, and do not consider any contribution from differences in motor unit size, recruitment or organization, some of which may in turn be related to handedness.

2.       TMS induces a complex series of unnatural, high-frequency discharges in the corticospinal system, and the pattern of these discharges is likely to be the major factor in determining the shape of the MEP. This should be discussed.

3.       The details of the experimental method are very sketchy. For example, were TMS stimuli delivered simultaneously to the two hemispheres or not? If not, were stimuli delivered in blocks or in an interleaved pattern? Was the timing of TMS randomized and unpredictable? What instructions were given to subjects? Assuming that the whole study was intended to be carried out in fully relaxed subjects, what controls were carried out to check for spontaneous ‘background’ EMG activity? Any such activity is likely to contribute very significantly to any trial-by-trial variability in MEP amplitude etc.

4.       The english in the Ms needs checking throughout.

Round 2

Reviewer 2 Report

Comments and Suggestions for Authors I have read the resubmitted version, and the authors have responded satisfactorily to all the points I raised...so no further criticisms.